# Food as Medicine: A Review of Plant Secondary Metabolites from Pollen, Nectar, and Resin with Health Benefits for Bees

**DOI:** 10.3390/insects16040414

**Published:** 2025-04-15

**Authors:** Bandele Morrison, Laura R. Newburn, Gordon Fitch

**Affiliations:** 1Canadian Hearing Services, Toronto, ON M5R 2V3, Canada; 2Centre for Bee Ecology, Evolution and Conservation, York University, Toronto, ON M3J 1P3, Canada; 3Department of Biology, York University, Toronto, ON M3J 1P3, Canada

**Keywords:** pollen, nectar, resin, honeybee, bumblebee, phytochemical, plant secondary metabolite, pathogen, parasite, pesticide, immune function, longevity

## Abstract

Currently, disease and exposure to pesticides are major conservation concerns in bees. A possible remedy to these stressors could be through plant products such as pollen, nectar, and resin. Specifically, products from various plants appear to improve bee health measures related to longevity, immune function, defense against disease, and detoxification of pesticides. This review provides an overview of ‘medicinal’ plant products relevant to wild bees (i.e., nectar, pollen, and resin), with a focus on the products’ chemical composition. Through a literature review, we identified 51 compounds with ‘medicinal’ benefits for bees in the pollen, nectar, and/or resin of over 200 species from 64 families. However, only a small number of these plant products have been directly tested on bees or their disease agents; in most cases, the benefit is inferred based on studies showing the effects of exposure to constituent chemical compounds. We point to future areas of multi-disciplinary investigation, including determining the degree of context- and dose-dependence in beneficial impacts, whether bees change their foraging behavior in response to disease, and how effective it would be to plant ‘medicinal’ plants near colonies suffering from disease and/or pesticide exposure.

## 1. Introduction

Most bees (Hymenoptera: Apoidea) rely exclusively on pollen and nectar for food. Pollen primarily serves as a source of protein and lipids, while nectar primarily provides carbohydrates. Generalist foragers collect pollen and nectar from multiple species to meet their dietary needs, as plant species differ in the nutrient profiles of their pollen and nectar [1,2,3,4,5,6,7,8,9,10,11,12]. However, nutrition is more than just macronutrients; in addition to carbohydrates, lipids, and proteins, the pollen and nectar of many species contain diverse plant secondary metabolites (PSMs, also known as phytochemicals). These compounds are typically thought to serve primarily defensive functions, protecting plants from herbivory, pathogens, and environmental stressors [13,14,15,16,17,18,19,20,21]. Despite their putative primary role as defensive compounds, recent research has revealed that a number of PSMs found in pollen and/or nectar are beneficial to pollinators, including through their anti-parasitic effects against bee endoparasites and pathogens [22] and detoxification of environmental toxicants [23,24,25].

These findings, coupled with widespread concern over declines in bee populations, driven in part by exposure to parasites, pathogens, and toxicants [26,27,28], have led to growing interest in leveraging ‘medicinal’ properties of bee-relevant floral products to promote bee health and conservation [29,30]. However, beyond a small number of well-studied species, we lack an understanding of how prevalent these ‘medicinal’ effects might be and how relevant they are to individual health and population dynamics in bees. This review seeks to synthesize the available information on the ‘medicinal’ effects of pollinator-relevant plant products (pollen and nectar) in order to begin to fill this knowledge gap. Given its use by honeybees to make propolis, an antimicrobial ‘bee glue’ used to line and fill cracks in the inner cavity of the hive, we also include examples of beneficial resins [31,32,33,34].

Evidence for plant products’ ‘medicinal’ effects on bees comes from multiple fields, including ecology, apiculture, veterinary sciences, and natural products medicine. A recent review [35] summarized the state of knowledge regarding the effectiveness of plant-derived products against honeybee pathogens, but they focused on veterinary treatments (i.e., essential oils), which tend to use highly concentrated plant extracts and do not reflect ecologically relevant exposures. We lack a comprehensive understanding of the current state of knowledge regarding the plant species whose pollen, nectar, and/or resin (PNR) are beneficial for pollinator health or the specific PSMs responsible for these benefits, particularly in an ecological context. Here, we address this gap with a review of the literature regarding the medicinal effects of plant products themselves on bees in ecologically relevant contexts. Specifically, we identify PSMs, as well as some fatty acids (FAs), and plant products containing these compounds, with beneficial effects on bee health. We focus on the eusocial honeybees (*Apis* spp. particularly *A. mellifera*) and bumblebees (*Bombus* spp.) because, to date, those taxa are the most researched; although, there is a bias toward *Apis* spp. given its use as a model organism for bee research in general. We hope this review will be useful in leveraging PSMs and other metabolites in PNR as a tool to control bee pathogens, spurring further research into the mechanisms underlying these effects and discovering heretofore undocumented ‘medicinal’ plant products.

## 2. Dimensions of Medicinal Benefits

In this review, we distinguish among multiple types of medicinal benefits: promoting longevity, reducing the impact of disease through immune function and defense against disease agents, and detoxifying toxicants. Below, we briefly define each benefit type in relation to honeybees and bumblebees.

### 2.1. Longevity

In honeybees, greater longevity is associated with higher reproductive success in queens [36,37]. For workers, earlier onset of foraging is correlated with a shorter lifespan; if stressors trigger precocious foraging, the number of workers may not be able to sustain the development of larvae, leading to colony death [38,39,40]. The relationship between longevity and colony success suggests that promoting factors that increase longevity could improve colony success in bees.

### 2.2. Immune Function and Protection Against Disease

Honeybees and bumblebees are susceptible to many disease agents, including bacteria, fungi, viruses, endoparasites, and ectoparasites [41,42,43]. While some of these disease agents are species-specific, there is increasing evidence that many can infect both honeybees and bumblebees [44,45,46]. Disease has been implicated as a driver of honeybee colony loss and bumblebee population declines [27,47,48,49,50], making it important to better understand the role of floral PSMs in influencing disease dynamics and health in pollinators. Plant secondary metabolites may influence disease in multiple ways, including via direct effects on the disease agent and through host-mediated effects [22]. Among host-mediated effects, perhaps the most significant is the potential for PSMs to influence the functioning of the bee immune system.

In eusocial bees, the immune response comprises both innate and social immunity. Innate immunity is the ensemble of cell signaling pathways that allow an individual bee to detect a disease agent and mount a response to neutralize it or repair harm caused by it [51,52,53,54]. Crucial elements in the bee immune system include the production of antimicrobial peptides (AMPs), melanization, and encapsulation [55,56]. Moreover, immune response pathways can be mediated by microbial symbionts [55,57], suggesting that the bee microbiome plays a key role in supporting immune function in eusocial bees. While details of the innate immune system may differ among bee taxa, the broad system is highly conserved across insects [52,54,58], meaning that observed effects of PSMs on immune function in one taxon may be more likely to apply across bee taxa.

Social immunity, by contrast, only operates in eusocial bees. It comprises a set of behaviors that help bees reduce the chance of infections and infestations, including auto- and allogrooming, removing corpses from the colony, and, for honeybees, collecting resin to make propolis [32,59,60,61,62].

### 2.3. Detoxification

Managed and wild bees are exposed to a variety of environmental toxicants, most significantly pesticides. These toxicants lead to acute and chronic effects, including reduced homing ability, foraging ability, cognitive function, and fecundity, and increased mortality [63,64,65,66,67,68,69,70,71,72,73,74,75,76,77,78]. Moreover, toxicant exposure often reinforces other stressors, ultimately leading to population decline [79]. For example, agrochemicals can interact with pathogens such that affected bees experience more negative outcomes when exposed to both [80,81,82,83,84]. Furthermore, exposure to pesticides has also been shown to reduce bee immune function, which can exacerbate existing infections [85,86,87,88]. The most commonly studied pesticides in bee research include fungicides (e.g., azoxystrobin, boscalid, propiconazole, and tebuconazole), pyrethroid insecticides (e.g., bifenthrin, cyfluthrin, and tau-fluvalinate), ryanoid insecticides (e.g., chlorantraniliprole), phenylpyrazole insecticides (e.g., fipronil) and neonicotinoid insecticides (e.g., imidacloprid, thiacloprid, and thiamethoxam) [89,90,91].

## 3. Plant Secondary Metabolites with Medicinal Benefits

From the literature, we identified 51 metabolites in pollen, nectar, and resin (PNR) with documented health benefits for bees (Appendix A; see Appendix A for details on literature review methods). These metabolites were distributed widely but unevenly across the plant phylogeny, with a total of 147 species comprising 64 families across 31 orders showing promise for medicinal effects (Figure 1A, Appendix A). We found evidence for the presence of medicinal metabolites in the flowers, but not explicitly in the floral products, of an additional 61 species, including 12 families that were otherwise not represented Appendix A. Beneficial metabolite diversity varied substantially, with Arecaceae, the highest-diversity family, having 14 documented beneficial metabolites from PNR, while 21 of the 64 families had a single beneficial metabolite (Figure 1B). Beneficial metabolites have, for the most part, not been documented from the more basal angiosperm orders; whether this reflects biological reality or simply sampling bias is not yet clear.

Eight beneficial metabolites were documented from ten or more families; these include quercetin (35 families in 22 orders), kaempferol (29 species, 19 orders), linolenic acid (22 families, 20 orders), linoleic acid (18 families, 16 orders), rhamnetin (15 families, 12 orders), p-coumaric acid (11 families, 8 orders), rutin (10 families, 10 orders), and cinnamic acid (10 families, 9 orders). Most families with high beneficial metabolite diversity (e.g., Arecaceae, Asteraceae, Brassicaceae, Fabaceae) share broadly similar suites of metabolites, including most of these widespread metabolites. Several families in the Lamiales (Lamiaceae and Verbenaceae) have very distinctive metabolite suites, which lack most of the widespread metabolites listed above and instead comprise a variety of terpenoids. Burseraceae, Rubiaceae, and Rutaceae are similarly distinctive, with low to moderate metabolite diversity but a high prevalence of terpenoids in the former and caffeine in the latter two (Figure 2).

The majority of records of beneficial metabolites in PNR come from pollen; we found a total of 301 records from pollen of 141 species, vs. 57 from nectar (of 31 species) and 65 from resin (of 28 species). Similarly, metabolite diversity was highest in pollen, though the difference here was not as substantial (35 different beneficial metabolites recorded from pollen, 28 from nectar, and 29 from resin). There is likely some sampling bias here, and we have little sense of the prevalence of negative results (i.e., of metabolites that were looked for but not detected), but nevertheless, these results are consistent with patterns of metabolite concentration across plant tissues, in which pollen typically has higher concentrations than nectar from the same species (e.g., [93,94,95]). Since pollen comprises the mobile gametes of plants (and as such, consumption of pollen can decrease plant fitness), while the role of nectar is exclusively to attract pollinators or natural enemies, higher abundance and diversity of metabolites in pollen vs. nectar is also consistent with theory and data that plants differentially allocate defensive compounds across tissue types depending on their importance to fitness (i.e., Optimal Defense Theory; [96,97,98]).

Where the effect of PNR or a constituent metabolite was evaluated in bees in vivo, *A. mellifera* was the most commonly studied species, with 72 distinct metabolites or plant products evaluated. For bumblebees, this number was 46, with nearly all these studies conducted on just two species: *B. impatiens* (38 metabolites or plant products) or *B. terrestris* (12 metabolites or plant products). Very few individual studies measured effects on multiple species (but see [99]), and only nine metabolites have been evaluated for medicinal effects in both *A. mellifera* and *Bombus* spp. However, the results are generally similar between the two taxa, with the degree of variation in results among studies of the same species equaling the variation between the two genera (this variation was often, though not always, due to contrasting effects on different target parasites or toxicants). This suggests that metabolite effectiveness may be generalizable between *Apis* and *Bombus* species. That said, one of the few studies that involved multiple species compared the effects of sunflower pollen on infection by *Crithidia bombi*, a trypanosomatid gut parasite, across multiple bumblebee species. It found species-specific responses to the plant product [99], arguing against generalizability. However, the mechanical basis for the anti-parasitic effect of sunflower ([100]; see Section 4.2 below) is distinctive. It may be that other kinds of effects are more generalized, but more studies that directly compare the effects of metabolites and/or PNR across multiple bee taxa are needed. Moreover, many of the most widespread metabolites, including kaempferol and p-coumaric acid, have only been evaluated in *A. mellifera*, while much of the research with *Bombus* has focused on more phylogenetically restricted metabolites, indicating an important knowledge gap.

Below, we summarize information regarding the phytochemicals with known beneficial effects on bees, distinguishing among four classes of benefits: longevity, immune function, defense against disease agents, and detoxification. However, we stress an important caveat: these benefits are frequently context-dependent, and consumption of a particular metabolite may confer a benefit in one of these dimensions but incur a cost along another dimension. Among the 51 metabolites included in this review, 76% had only documented positive effects on one or more dimensions of bee health, and an additional 24% had context-dependent effects. None of the reported metabolites in this review had purely negative effects. A thorough exploration of the potential negative impacts of metabolites on bee health is beyond the scope of this review but clearly warrants further study.

### 3.1. Longevity

Only a few metabolites have been associated with increased longevity in bees: abscisic acid, caffeine, resveratrol, and thymol [101,102]. Furthermore, the effects of abscisic acid appear to depend on the focal level of organization, reducing individual longevity but promoting colony longevity in honeybees [101]. By contrast, multiple metabolites with beneficial impacts in another dimension of health are associated with decreases in individual longevity in at least some contexts; these include acetic acid [103], eugenol [104], p-coumaric acid [103], and thymol [104]. These findings highlight how a single metabolite can influence multiple health parameters, sometimes in opposing ways, making it difficult to make blanket generalizations about the benefit or detriment of specific metabolites without reference to specific context.

### 3.2. Immune Function and Microbiome

A wider variety of metabolites have been shown to positively influence bee immune function and/or microbiome, though most studies to date have been performed in honeybees (a recent study with *Bombus impatiens* found no effect of pollen diet on melanization, a measure of the immune response, comparing three monofloral pollen diets and diets including two or three pollen species [105]). Immune gene expression and antimicrobial peptide production in honeybees are enhanced by p-coumaric acid, anabasine, aucubin, catalpol, eugenol, and nicotine [104,106,107]. Abscisic acid increases granulocyte activation, which contributes to wound healing [108,109], and caffeine, gallic acid, kaempferol, and p-coumaric acid increase the diversity, abundance, and richness of the gut microbiome [110]. By contrast, cinnamic acid and hesperidin, despite having anti-fungal effects (Table 1), are associated with reduced expression of the key immune gene hymenoptaecin [111].

Of the metabolites associated with increased immune function, kaempferol, abscisic acid, gallic acid, and p-coumaric acid have been found in the PNR of multiple families that are not closely related, suggesting they may be widespread. Similarly, eugenol has been documented in multiple families. Caffeine, an alkaloid, has been documented in a smaller number of families, most notably Rubiaceae and Rutaceae. Anabasine and the iridoid glycosides aucubin and catalpol are more restricted still, with documented occurrence in *Nicotiana* spp. (Solanaceae) and several families in the Asterids, respectively (Figure 2).

### 3.3. Defense Against Disease Agents

By far, the most documented benefit from metabolites is defense against disease agents, with 41 of the total 51 beneficial metabolites showing defensive effects against one or more disease agents (Table 1). The bulk of ecological studies have been conducted on trypanosomatid and microsporidian parasites, while apicultural research has focused more on bacteria and *Varroa* mites. Given differences in emphasis and methodology between these fields, caution is warranted in comparing the resulting data. For example, to date, research on pathogenic bacteria such as *Paenibacillus* spp. has been primarily conducted on cultures of the disease agent, while research on microsporidians and trypanosomatids has been conducted in multiple contexts, but with an emphasis on in vivo studies of previously infected individuals. However, the effects of metabolites observed in vitro often do not translate fully to effects in the host (e.g., [112]), making it difficult to generalize the effects of a metabolite based on a single exposure route. There is mounting evidence that metabolites are transformed during passage through the bee gut, either via direct effects of metabolic processes in the bee or mediated by gut microbiome [113,114]. Host-mediated transformation of metabolites also explains, at least in part, why some compounds have prophylactic but not therapeutic effects (e.g., callunene against *C. bombi* in *Bombus terrestris* [113]) or vice versa (e.g., tiliaside against *C. bombi* in *Bombus terrestris* [114]). While effects documented from in vitro studies represent direct effects of the compound on the disease agent, most studies conducted in vivo have not identified the causal mechanism linking the metabolite or plant product to reduced infection (but see [100,113]), meaning these could represent either direct effects or host-mediated effects (e.g., by improving immune function).

### 3.4. Detoxification

We found evidence of detoxifying effects for 11 metabolites (Table 2). Some of these metabolites (e.g., quercetin) appear to help bees by increasing the expression of detoxification genes [106,115]; the mode of action for other detoxifying metabolites is not clear. Furthermore, there are multiple metabolites in this group that appear to have context-dependent effects. For example, luteolin appears to worsen the effects of tebuconazole but lessens the effects of tau-fluvalinate on honeybees [24]. Another metabolite with context-dependent effects in honeybees is indole-3-acetic acid in honeybees: at a high concentration, it causes a reduction in survival, while at an intermediate concentration, it causes an increase in survival [116].

## 4. Considerations and Future Directions

Despite growing attention to the topic, our understanding of the metabolite content of PNR and its relevance to bee health is quite limited. Most studies investigating metabolite content focus on leaves, seeds, and/or roots of plants, often with the intent of using the findings to advance human health and/or agricultural productivity [117,118,119,120,121,122,123,124]. In comparison, relatively few studies investigate the metabolites in pollinator-relevant plant products. More investigation, including investigations into clades that have, to date, been underrepresented in sampling, is required to improve our understanding of the metabolites present in PNR and how they influence bees. In addition, for plants with flowers known to contain beneficial metabolites Appendix A, the specific parts of the flower that confer benefits should be identified. A recent study that systematically analyzed the nutrient profile of nectar and pollen of North American plants [125] could serve as a model for the systematic investigation of the metabolite content of PNR and identify other plant species that may provide medicinal benefits to bees.

Beyond this relatively straightforward question of what metabolites are present in PNR and how they influence bee health, there are several more complicated questions, particularly relating to the application of this information to bee conservation efforts.

### 4.1. Complications with Inferring Benefit from Chemistry

As promising as medicinal metabolites and plant products may seem in alleviating the effects of disease and pesticide exposure in bees, there are many cases where the quantity makes a difference. For example, despite the medicinal benefits of consuming sunflower pollen, *B. impatiens* workers fed pollen exclusively from sunflower experienced increased mortality relative to other monofloral or polyfloral pollen diets [126]. In another example, a low concentration of anabasine in artificial nectar solutions had no effect on *C. bombi* infection in *B. impatiens*, but higher concentrations significantly reduced *C. bombi* infection [127]. These instances highlight the necessity of identifying thresholds for positive and negative effects of metabolites and plant products in bees. Experiments that evaluate the effects of multiple concentrations of a putatively beneficial metabolite (e.g., [94,126]), combined with quantification of its concentration in relevant PNR, will be needed to predict the health impact of consumption of this product.

Moreover, documenting the presence of a beneficial metabolite in the PNR of a particular plant species does not necessarily mean that this benefit will accrue to a bee exploiting that resource. There are several complicating factors. First, many metabolites display context-dependent effects. Such context-dependency can arise from many sources, including synergistic or antagonistic interactions among metabolites [128,129], the nutritional or health status of the bee [130,131], and environmental context [132]. Second, metabolite content varies substantially both within individuals and among individuals of the same species due to, among other factors, life stage, herbivore pressure, and abiotic conditions (particularly nutrient availability) [133,134]. While this variation is well documented among vegetative tissues, the degree of intraspecific variation in metabolite content of PNR is less well known. In one of the few studies to investigate intraspecific variation in pollen and nectar metabolite content, there was a significant variation both within and between populations and cultivars for most of the >30 species sampled, suggesting that variability is the norm [135]. Since, as previously noted, the beneficial effects of metabolites are often highly dependent on exposure dose, this outcome suggests that context dependency may rule the day in determining the health outcome of consumption of plant products that have proven beneficial in some contexts.

### 4.2. Medicinal Benefits of Pollen, Nectar, and Resin Beyond Phytochemistry

While the focus of this review is on the health benefits of metabolites found in PNR, other features beyond PNR chemistry may contribute to their beneficial impacts. One particularly salient example is the potential mechanical effects of spiky pollen. The pollen of *Helianthus annuus* (common sunflower) and related species dramatically reduces *C. bombi* infection in *B. impatiens* workers [136,137,138,139,140,141,142,143]. However, despite an array of potentially beneficial metabolites in *H. annuus* pollen Appendix A, this effect does not stem from the metabolite content [139]. Rather, spines on the pollen exine appear to mechanically inhibit colonization of the hindgut lining by *C. bombi* [100]. These findings highlight the need to carefully consider mechanisms when linking the consumption of PNR (as opposed to specific metabolites) to health impacts. On the other hand, while studies that test the effects of specific metabolites make it easier to infer mechanisms, they overlook the complex ways in which multiple metabolites may interact with one another.

### 4.3. To What Extent Is PSM Content Driving Bee Foraging Behavior?

Bees are attracted to flowers based on multiple traits, including their macronutrient profile; morphological traits, including shape and color; and abundance [3,144,145,146,147,148,149]. A relatively small number of studies generally focused on plants’ highly distinctive PSM profiles (in particular, with PNR that include compounds that are toxic at high but ecologically relevant doses) have demonstrated that both bumblebees and honeybees discriminate among food sources on the basis of PSM content [150,151,152,153] and that this may depend on the health status of individual bees [94,154] or the colony [155]. However, whether PSM content is a trait driving pollinator foraging choices in less extreme cases is largely unknown. To address this knowledge gap, lab studies should evaluate bees’ abilities to detect PSMs (particularly those with weaker beneficial effects). We suspect that in most cases, PSM content is less important than macronutrients or morphological traits in driving pollinator attraction, but this area should be more rigorously tested.

### 4.4. Applications

A key proposed intervention for bee conservation has been increasing the abundance of ’pollinator friendly’ plants, but this intervention is not sufficient to address the non-nutrition-related stressors affecting bees [156,157,158], including exposure to pesticides and pathogen spill-over. Given the growing body of evidence on the ’medicinal’ value of PNR from some species, certain researchers have recommended selectively including these species in ‘pollinator-friendly’ plantings [110,159]. However, there are many questions regarding the effectiveness of this approach that should be answered before it is recommended as a conservation strategy. First, as noted above, the extent to which bees make foraging choices based on PSM content is not clear. While preferential feeding on plants with beneficial PSMs (i.e., self-medication) is not strictly necessary for the selective planting of these species to be beneficial to bee health, it would make such efforts more likely to succeed. Second, in the case of antiparasitic effects, evidence from lab studies indicates that repeated exposure to PSMs at ecologically relevant concentrations can lead to parasite resistance [160]. While the evolution of resistance is less likely in natural conditions, where PSM exposure is likely to be more variable through time, we need further investigation into the potential for parasites to develop resistance. Obviously, if the evolution of resistance occurs in natural conditions, this would sharply limit the long-term effectiveness of ’medicinal’ plantings. Ultimately, we need more field experiments that track the effects of increasing the abundance of ‘medicinal’ plants on both bee colony health measures and population dynamics over time [161].

## 5. Conclusions

Overall, there is strong evidence for plants with medicinal benefits for bees. However, of the >200 ‘medicinal’ plants featured in this review, only a small fraction provides bee-relevant products that have been tested directly in bees or their disease agents. The small percentage demonstrates the need for additional investigation into the effects of pollen, nectar, and resin on bee health outside of nutrition.

## Figures and Tables

**Figure 1 insects-16-00414-f001:**
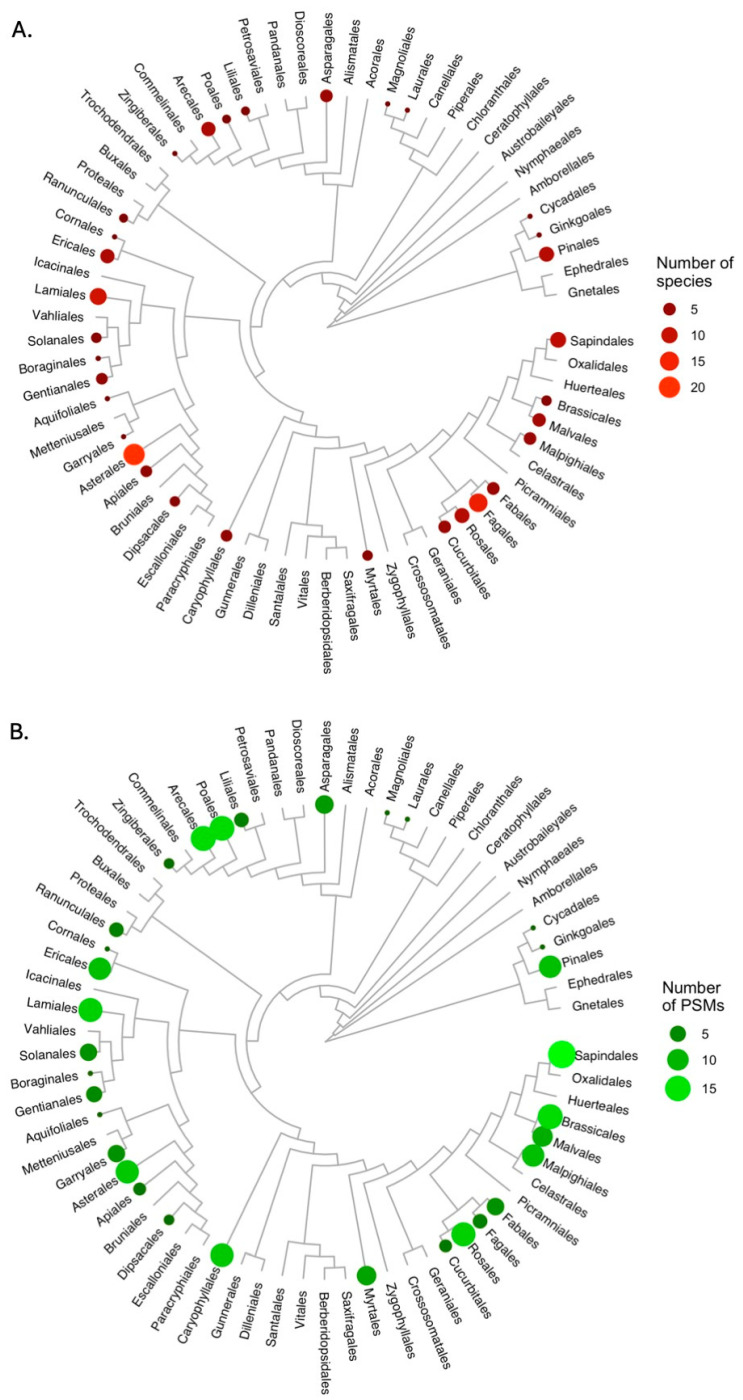
Phylogenetic distribution of medicinal plants for bees. (**A**) Plant species whose floral products and/or resin have documented health benefits for bees or contain metabolites with medicinal benefits for bees. (**B**) metabolites with medicinal benefits. Phylogeny is based on [92].

**Figure 2 insects-16-00414-f002:**
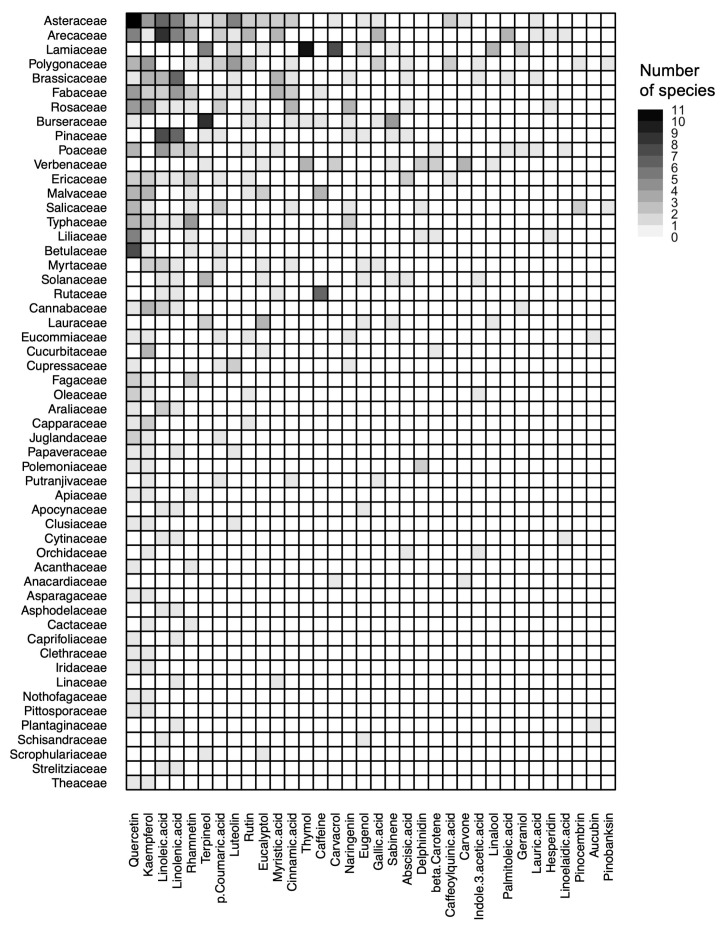
Adjacency matrix indicating the distribution of beneficial metabolites (columns) across plant families (rows). Only metabolites occurring in >1 family and families with >1 documented metabolite are included. Shading in each square indicates the number of species in that family whose pollen, nectar, and/or resin are known to contain the relevant metabolite.

**Table 1 insects-16-00414-t001:** Metabolites found in nectar, pollen, and/or resin with documented effectiveness against disease agents in honeybees and/or bumblebees.

Disease Agent	Compound	Effect Type ^†^	Representative Species
Viruses	Anabasine	Therapeutic	*Nicotiana* spp.
	Caffeine	Therapeutic	*Coffea* spp. *Citrus* spp. *Onobrychis vicifolia*
	Catalpol	Therapeutic	*Chelone glabra*
	Hesperidin	Therapeutic	*Prunus cerasus*
	Nicotine	Prophylactic	*Nicotiana* spp.
Bacteria	Carvacrol	Culture	*Origanum vulgare*
	Eicosenoic acid	Culture	*Brassica napus*
	Hexenal	Culture	*Citrus maxima*
	Hyperforin	Culture	*Hypericum* spp.
	Lauric acid	Culture	*Brassica napus*
	Linoelaidic acid	Culture	*Zea mays*
	Linoleic acid	Culture	Widespread
	Linolenic acid	Culture	Widespread
	Myristic acid	Culture	*Helianthus annuus*
	Palmitoleic acid	Culture	*Brassica napus*
	Pinobanksin	Culture	*Populus nigra*
	Pinocembrin	Culture	*Fagopyrum esculentum*
	Rhamnetin	Culture	*Quercus acutissima*
	Sabinene	Culture	*Thymus vulgaris*
	Terpeniol	Culture	*Thymus vulgaris*
	Thymol	Culture	*Thymus vulgaris*
	Tridecanoic acid	Culture	*Zea mays*
	Undecanoic acid	Culture	*Zea mays*
*Ascosphaera apis*	Cinnamic acid	Culture	*Prunus cerasus*, *Taraxacum* spp.
	Pinobanksin	Culture	*Populus nigra*
	Pinocembrin	Culture	*Fagopyrum esculentum*
Microsporidians (*Vairimorpha* and *Nosema*)	Abscisic acid	Therapeutic	*Arbutus unedo*
	Acetic acid	Therapeutic	*Acer saccharum*
	Caffeine	Prophylactic, Therapeutic	*Coffea* spp. *Citrus* spp. *Onobrychis vicifolia*
	Gallic acid	Therapeutic	*Phoenix dactylifera*, *Fagopyrum esculentum*
	Kaempferol	Therapeutic	Widespread
	Naringenin	Therapeutic	*Brassica napus*, *Prunus avium*
	p-Coumaric acid	Therapeutic	Widespread
Trypanosomatids (*Crithidia* and *Lotmaria*)	Anabasine	Therapeutic	*Nicotiana* spp.
	Aucubin	Culture	*Chelone glabra*
	Caffeoylquinic acid	Therapeutic	*Carthamus tinctorius*, *Fagopyrum esculentum*
	Callunene	Prophylactic	*Calluna vulgaris*
	Carvacrol	Therapeutic	*Origanum vulgare*
	Catalpol	Therapeutic	*Chelone glabra*
	Cinnamaldehyde	Therapeutic	*Schisandra chinensis*
	Eugenol	Therapeutic	*Ocimum basilicum*, *Laurus nobilis*
	Gelsemine	Therapeutic	*Gelsemium sempervirens*
	Nicotine	Therapeutic	*Nicotiana* spp.
	Thymol	Therapeutic	*Thymus vulgaris*
*Varroa*	Abscisic acid	Therapeutic	*Arbutus unedo*
	Carvone	Therapeutic	*Artemesia fragrans*
	Eugenol	Therapeutic	*Ocimum basilicum*, *Laurus nobilis*
Small hive beetle (*Aethina tumida*)	Acetic acid	Therapeutic	*Acer saccharum*

^†^ Therapeutic: reduces infection intensity when consumed by already-infected individuals; prophylactic: reduces likelihood of infection or subsequent infection intensity when consumed by uninfected individuals prior to exposure; culture: direct exposure in vitro reduces density and/or viability of disease agent.

**Table 2 insects-16-00414-t002:** Plant secondary metabolites with detoxifying effects for bumblebees and honeybees exposed to pesticides. The metabolites listed in this table are found in the pollen, nectar, and/or resin of plants featured in this review.

Compound	Pesticide	Representative Species
Abscisic acid	Imidacloprid	*Solanum lycopersicum*
Caffeine	Thiamethoxam	*Coffea* spp.
Gallic acid	Thiamethoxam	*Corymbia citriodora*
Indole-3-acetic acid	Tau-fluvalinate	*Olea europea*
Kaempferol	Thiamethoxam, imidacloprid	Widespread
Luteolin	Tau-fluvalinate	*Allium cepa*
Myristic acid	Tebuconazole	*Linum usitassimum*
p-Coumaric acid	Chlorantraniliprole, cyfluthrin, imidacloprid, propiconazole, tau-fluvalinate, thiamethoxam	Widespread
Quercetin	Bifenthrin, boscalid, chlorantraniliprole, imidacloprid, propiconazole, tau-fluvalinate, tebuconazole	*Anethum graveolens*
Rutin	Fipronil, imidacloprid	*Tilia* spp.

## Data Availability

No data were produced as part of this study.

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
