# Peer review of "Food as Medicine: A Review of Plant Secondary Metabolites from Pollen, Nectar, and Resin with Health Benefits for Bees"

_insects, 2025, doi:10.3390/insects16040414_

Round 1
Reviewer 1 Report
Comments and Suggestions for Authors
Pollination by honeybee is indispensable for maintaining the ecological balance and agricultural production. But honeybee’s population has been declining for two decades, bring seriously effects on food provision. The health and population of honeybee attract the attention all over the world. Bees rely on pollen and nectar for nutrition. In past, there were some studies focused on the effect of plant species or diversity of pollen or honey on the macronutrients and honeybee health. While in this review, the authors address the medical role of plant secondary metabolites from honey, pollen and resin, such as promoting longevity and immune function, protecting against disease agents, and detoxifying toxicants. They referred many literature from different disciplines and fields, presenting the PSMs with health benefits, and the plant species whose floral products and/or resin containing them. The review bring the PSMs to the field of bee health, thereby is scientific sense. Here are some suggestions:
- I understand that, in response to honeybee population decline, the authors pay attention to the health benefits of plant secondary metabolites from pollen, nectar and resinin the review. Effect of PSMs on honeybee was relative a new field in bee science, the review would certainly attract more attention on PSMs to bee health, which was the just scientific point of the review. However, PSMs was not always beneficial to bees, part of them might be harmful. If these adversary effect of PSMs were also involved in the review, that will make the review more comprehensive and more valuable.
- If the first suggestion was acceptable, the title of the review should be revised accordingly.
- The format of the review is a little strange. It seems not a like typical review, since it contain the “Research methods”and “Results and Discussion” part, that make it more like a research article. It is suggested to reorganize the manuscript according to the form of a typical review. “Research methods” part could be provided in supplementary files.
- In line 86-149, the author interpreted the “Dimensions of medicinal benefits”of Longevity, Immune function and protection against disease and Detoxification. These conceptions were regular content, easy to understand and constitute the main parts in “Results and Discussion” . Thereby, I think the “Dimensions of medicinal benefits” part is not necessary to be kept here, which could be merged with the content in introduction or discussion part.
- Literature citation error, such as in line 374:LoCascio, Aguirre, et al. 2019; line 375: Fowler, Sadd, et al.; line 416: (Palmer-Young, B. M. Sadd, et al. 2017). Please check throughly and revised it.
Reviewer 2 Report
Comments and Suggestions for Authors
The manuscript by Morrison et al. is a comprehensively written review of plant natural products beneficial for bees. I enjoyed reading it and have no major concerns, only a few minor suggestions, as follows:
- line 73: move `(2023)` up to line 70 after `Bava and colleagues`
- line 93: `For workers, age at onset of foraging is correlated with age at death` - please clarify
- Figure 2: add scale for grey shading
- line 228: define `C. bombi` at first mention
Round 2
Reviewer 1 Report
Comments and Suggestions for Authors
Based on the comments and suggestions given to the last version of manuscript, the literature citation error was corrected, and the format of review was also adjusted accordingly in this version. For the other comments and suggestions, the author gave detail explainations. I think it is acceptable now for publishin in the journal.